# A Family of Externally-Functionalised Coordination Cages

Garrett D. Jackson, Max B. Tipping, Christopher G. P. Taylor , Jerico R. Piper, Callum Pritchard, Cristina Mozaceanu and Michael D. Ward *

Department of Chemistry, University of Warwick, Coventry CV4 7AL, UK;
Garrett.Jackson@warwick.ac.uk (G.D.J.); M.Tipping@warwick.ac.uk (M.B.T.);
C.Taylor.10@warwick.ac.uk (C.G.P.T.); Jerico.Piper@warwick.ac.uk (J.R.P.);
Callum.Pritchard@warwick.ac.uk (C.P.); Cristina.Mozaceanu@warwick.ac.uk (C.M.)
* Correspondence: m.d.ward@warwick.ac.uk

**Abstract:** New synthetic routes are presented to derivatives of a (known) $M_8L_{12}$ cubic coordination cage in which a range of different substituents are attached at the $C^4$ position of the pyridyl rings at either end of the bis(pyrazolyl-pyridine) bridging ligands. The substituents are (i) –CN groups (new ligand $L^{CN}$), (ii) –$CH_2OCH_2$–CCH (containing a terminal alkyne) groups (new ligand $L^{CC}$); and (iii) –$(CH_2OCH_2)_3CH_2OMe$ (tri-ethyleneglycol monomethyl ether) groups (new ligand $L^{PEG}$). The resulting functionalised ligands combine with $M^{2+}$ ions (particularly $Co^{2+}$, $Ni^{2+}$, $Cd^{2+}$) to give isostructural $[M_8L_{12}]^{16+}$ cage cores bearing 24 external functional groups; the cages based on $L^{CN}$ (with $M^{2+} = Cd^{2+}$) and $L^{CC}$ (with $M^{2+} = Ni^{2+}$) have been crystallographically characterised. The value of these is twofold: (i) exterior nitrile or alkene substituents can provide a basis for further synthetic opportunities via 'Click' reactions allowing in principle a diverse range of functionalisation of the cage exterior surface; (ii) the exterior –$(CH_2OCH_2)_3CH_2OMe$ groups substantially increase cage solubility in both water and in organic solvents, allowing binding constants of cavity-binding guests to be measured under an increased range of conditions.

**Keywords:** coordination cage; crystal structure; host-guest chemistry



## 1. Introduction

The ability of self-assembled coordination cages—hollow, pseudo-spherical, metal-ligand assemblies [1–7]—to bind small-molecule guests in their central cavities has resulted in a wide range of potential applications such as transport and release of 'cargoes' [8–11] including drug molecules; catalysed reactions of cavity-bound guests whose reactivity is altered [12–21]; and analysis or sensing of species whose binding in the cavity triggers an optical response [22]. Guests bound inside coordination cages span a huge range from simple anions [23] or solvent molecules [24] via fullerenes [25,26] to small proteins [27,28].

Accordingly, the main focus on coordination cage chemistry has been guest binding in the central cavity which is now a very well-developed area. Rather less attention however has been given to the functionalisation of the exterior surfaces which can be an equally important aspect of cage chemistry. Carefully chosen substituents attached to cage exterior surfaces can control solubility [29,30]; provide functional groups which can interact with surfaces [31] or proteins [32]; and provide functionality such as redox [33] or photophysical [34] properties that supplement the properties of the cage/guest assembly.

In this paper, we report a series of synthetic studies on our octanuclear cubic coordination cage assembly **H** (Figure 1) [35,36] which are focused on the exterior surface. The parent cage **H** (and structurally equivalent complexes with other metal ions at the vertices) as its fluoroborate salt is soluble in polar organic solvents, in which early studies of guest binding were performed [29,37]; the addition of hydroxymethyl substituents to the exterior surface to give **H$^w$** (Figure 1) improved water solubility [29], allowing for a much stronger binding of a wide range of neutral organic guests because of the magnitude

of the hydrophobic effect which dominates guest binding in aqueous solution [24]. However, the inclusion of the hydroxymethyl substituent at the $C^4$ position of each pyridine ring substantially complicated the ligand synthesis, limiting our ability to apply the same methodology to other external functional groups.

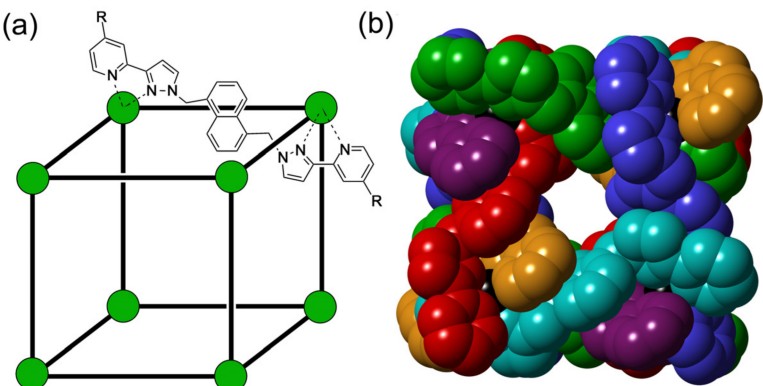

**Figure 1.** (**a**) A sketch of the cubic host cage $[M_8L_{12}]^{16+}$, abbreviated as **H** (R = H), emphasising the cubic array of Co(II) ions and the disposition of one bridging ligand; and its derivatives bearing substituents at the 24 externally-directed pyridyl $C^4$ positions **$H^W$** (R = $CH_2OH$), **$H^{CN}$** (R = CN), **$H^{CC}$** (R = $CH_2OCH_2$-CCH) and **$H^{PEG}$** [R = –$(CH_2OCH_2)_3CH_2OMe$]. (**b**) A space-filling view of the core showing the intertwined ligand array, with each ligand coloured separately for clarity.

Accordingly, we have improved the synthetic procedure associated with the ligands which form the basis of cages of the **H** family, to provide a more general route to a wider variety of externally-functionalised cages. As well as allowing substantially improved solubility of the cage family by allowing the straightforward inclusion of external solubilising substituents, we have incorporated as external functional groups both alkyne and nitrile substituents, which provide a useful basis for elaboration into a wide range of other functional groups; structurally characterised examples of these alkyne-substituted and nitrile-substituted analogues of **H** are presented. In short, this synthetic study greatly improves the possibilities for external functionalisation of the cage family with a wide variety of different substituents which are of value for a range of purposes.

## 2. Materials and Methods

### 2.1. General Details

The ligand **$H^W$** was prepared as previously reported by us [29]; 2-acetyl-4-cyanopyridine was prepared according to the literature method [38]. Other organic reagents and metal salts were purchased from Sigma-Aldrich and used as received. Instrumentation used for routine spectroscopic measurements was as follows: $^1$H-NMR spectroscopy, Bruker Avance 300, 400 or 500 MHz instruments; ES mass spectrometry, a Bruker Compact ESI-Q-TOF in positive ion mode; fluorescence spectra, an Agilent Cary Eclipse instrument. Details of the synthesis and characterisation of new ligands and complexes are in the Supporting Information.

### 2.2. X-ray Crystallography

The X-ray crystallographic data for the structure determination of **$H^{CC}$** were collected in-house using a Rigaku Oxford Diffraction Synergy S with a HyPix-6000HE detector; data for the structure determination of **$H^{CN}$** were collected in Experiment Hutch 1 of beamline I–19 at the UK Diamond Light Source synchrotron facility [39]. Details of software used for structure solution and refinement have been previously reported [40]. Crystallographic, data collection and refinement parameters are collected in Table 1. CCDC deposition numbers: 2110709–2110710. In both cases, as is usual for large supramolecular assemblies of this type, a combination of weak scattering associated principally with the disorder

of anions and solvent molecules required extensive use of geometric restraints during refinement to ensure a physically meaningful and stable refinement. Diffuse electron density that could not be modelled satisfactorily was removed from the refinement using the solvent mask feature in OLEX; details are in the individual CIFs.

**Table 1.** Summary of crystallographic, data collection and refinement parameters for the two crystal structures in this paper.

| Compound | Cd•H$^{CN}$(•10.5MeCN) | Ni•H$^{CC}$(•20DMF) |
|---|---|---|
| Empirical formula | $B_{16}C_{381}Cd_8F_{64}H_{271.5}N_{106.5}$ | $B_{16}C_{492}F_{64}H_{496}N_{92}Ni_8O_{44}$ |
| Formula weight | 8629.69 | 9310.18 |
| $T$/K | 100(1) | 100(1) |
| Crystal system | Triclinic | Monoclinic |
| Space group | $P$–1 | $P2_1/n$ |
| Crystal size / mm$^3$ | $0.08 \times 0.03 \times 0.02$ | $0.25 \times 0.20 \times 0.15$ |
| $a$/Å | 21.1212(6) | 23.52269(18) |
| $b$/Å | 22.2410(9) | 46.1903(5) |
| $c$/Å | 24.6693(11) | 23.79481(19) |
| $\alpha$/degrees | 116.279(3) | 90 |
| $\beta$/degrees | 98.215(3) | 101.6645(8) |
| $\gamma$/degrees | 101.119(3) | 90 |
| $V$/Å$^3$ | 9846.7(4) | 25319.6(4) |
| Z | 1 | 2 |
| $\rho_{calc}$/g cm$^{-3}$ | 1.455 | 1.221 |
| $\mu$/mm$^{-1}$ | 0.485 | 1.064 |
| Radiation | Synchrotron ($\lambda = 0.6889$) | CuK$\alpha$ ($\lambda = 1.54184$) |
| Reflections collected | 122279 | 541416 |
| Data/restraints/parameters | 39594/8799/2566 | 54537/6902/2953 |
| Final $R$ indexes [$I \geq 2\sigma(I)$] | $R_1 = 0.0900$, w$R_2 = 0.2744$ | $R_1 = 0.1412$, w$R_2 = 0.3935$ |
| Final $R$ indexes (all data) | $R_1 = 0.1528$, w$R_2 = 0.3222$ | $R_1 = 0.1694$, w$R_2 = 0.4268$ |

## 3. Results

### 3.1. Reactions to Introduce 2-Acetyl Group onto Pyridine Nucleus

The original simple synthesis of the ligand L used for assembly of **H** [35] is based on the easy availability of 2-acetylpyridine, whose acetyl group is trivially converted via two simple steps to a pyrazole group [41,42], furnishing the 3-(2-pyridyl)-pyrazole unit which is the essential chelating unit of all ligands in this cage family. The adaptation of this synthesis to incorporate a substituent X at the pyridine C$^4$ position immediately presents a greater challenge as the desired 4-X-2-acetylpyridines are rarely commercially available. To make the hydroxymethyl-substituted ligand **L$^w$** (used as the basis of the water-soluble cage **H$^w$**) [29] we started with 4-hydroxymethylpyridine, protected the HO group by silylation, and then introduced the 2-acetyl group in a 3-step sequence involving conversion of the pyridine to a pyridine-*N*-oxide, reaction with a cyanide source to give the 2-cyanopyridine, and then a Grignard reaction with MeMgI to give the necessary 2-acetylpyridine bearing a protected hydroxymethyl substituent. This worked but is rather cumbersome and the final Grignard reaction to generate the acetyl group is low-yielding [29].

A one-step procedure to introduce the 2-acetyl group to the pyridine nucleus, which is much quicker and higher-yielding and which is also tolerant of a range of substituents at the C$^4$ position of the pyridine ring, is the reaction shown in Scheme 1, reported in 1991

by Fontana and co-workers, which involves the in situ generation of an acyl radical by decarboxylation of pyruvic acid using persulfate and an Ag(I) catalyst [38]. It will be clear that this is immediately both simpler and more versatile and has become our default route to synthesise ligands of this type. It opens up the chemistry of externally-substituted cages to a wide range of functional groups, as we show below for the preparation of **L^PEG** and **L^CN** and their associated cage complexes.

**Scheme 1.** One-step introduction of a 2-acetyl group onto a 4-substituted pyridine according to ref. [38]: (i) pyruvic acid, $(NH_4)_2S_2O_8$, catalytic $AgNO_3$. The subsequent sequence of steps for conversion of the 2-acetyl group to a pyrazole, and thence to the completed ligands **L^R**, have been reported previously [29,35].

### 3.2. Preparation of L^PEG and the Associated Cage H^PEG

A particularly useful set of external substituents on the cage core is the poly(ethyleneglycol) (PEG) chains, giving the ligand **L^PEG** which in turn forms the cage **H^PEG**. The starting material 4-hydroxymethylpyridine was first alkylated with the tosylate of tri-ethyleneglycol monomethyl ether (Supplementary Materials, p. 2) to give the PEG-ylated pyridine which could then be converted to the 2-acetyl-4-PEG-pyridine using the method of Scheme 1. The remaining steps to convert the acetyl group to a pyrazole [41,42], and then connect two of the PEG-ylated 3-(2-pyridyl)pyrazole units to the central naphthalene-1,5-diyl core to give **L^PEG** [29,35], were carried out following the general methodology reported earlier (Supplementary Materials, p. 3–6): variations in workup associated with the presence of the PEG chains (which require, for example, different conditions for chromatography) are required.

The reaction of **L^PEG** with $Co(BF_4)_2$ or $Cd(NO_3)_2$ in the required 3:2 molar ratio in methanolic solution at 60 °C afforded the PEG-ylated cages **Co•H^PEG** and **Cd•H^PEG**, as confirmed by $^1$H NMR spectroscopy and ES mass spectrometry (Supplementary Materials, p. 7–8), with the ES mass spectra showing a characteristic sequence of peaks for the intact cage cation associated with varying numbers of counter-ions. The addition of the PEG substituents results in complexes that clearly tumble slowly in solution, giving particularly broad $^1$H NMR signals (in addition to the inherent paramagnetic broadening associated with **Co•H^PEG**) so individual peaks are not assignable: but the number of signals is consistent with expectations based on the symmetry of the cages with two independent ligand environments [29,35]. These PEG-ylated cages are much more soluble in water than **H^W**, and also much more soluble in organic solvents (including low-polarity ones like $CHCl_3$) than unsubstituted **H**. This apparent dichotomy arises from the flexibility of PEG chains which have both polar and non-polar conformations according to how they are folded [43,44].

This substantially improved the solubility of the host in $CHCl_3$ and has immediate benefits in terms of improved binding in the cavity of H-bond accepting guests such as members of the coumarin family, which interact with an H-bond donor site on the cage interior surface; this cage/guest H-bonding interaction has been analysed in detail before [29,37], and the nature of the interaction has been confirmed by X-ray crystallography of cage/guest complexes [40]. Several 1:1 binding constants of substituted coumarins inside the cavity of **H** were previously measured in MeCN and observed to have *K* values of the order of $10^2$ M$^{-1}$ based on this interaction [29,37]. We can immediately see the benefit of a solvent such as $CHCl_3$ that is a poorer competitor for hydrogen-bonding sites: for example, the binding constant of 4-methyl-7-aminocoumarin (MAC) in the cavity of **Co•H^PEG** in

CHCl₃, measured by the progressive quenching of MAC fluorescence on the addition of portions of **Co•H$^{PEG}$** (Figure 2), is $3.8 \times 10^4$ M$^{-1}$, an increase in $K$ of *ca.* 2 orders of magnitude compared to what we have previously observed for coumarin derivatives in MeCN [37]. Given how much our previous studies of assemblies based on **H** and **H$^w$**, from the measurement of guest binding constants [29,36] to the study of photoinduced electron transfer in cage/guest assemblies [45,46], is limited by solubility issues the combination of: (i) greatly increased solubility in both water and organic solvents and; (ii) stronger binding of hydrogen-bonding guests in organic solvents provided by the **H$^{PEG}$** complexes, will greatly facilitate future studies on properties of cage/guest assemblies.

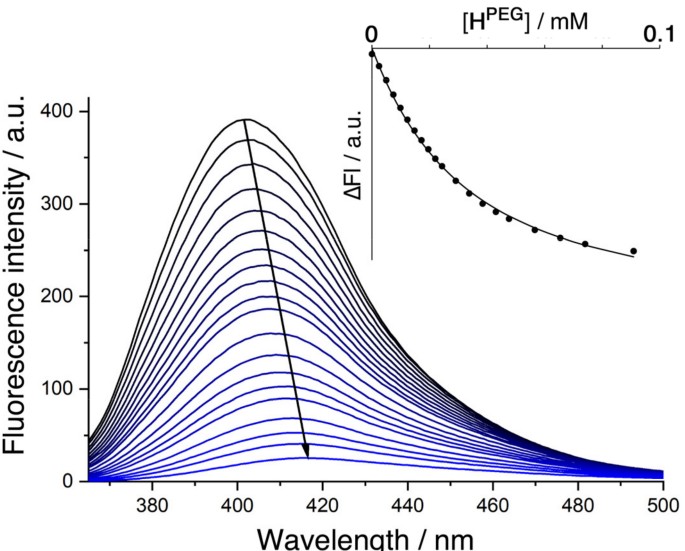

**Figure 2.** Fluorescence quenching of 4-methyl-7-aminocoumarin ($10^{-5}$ M in CHCl₃, 293 K) on the addition of increasing amounts of **Co•H$^{PEG}$** (up to 9 equivalents); the calculated binding constant (see main text) arises from fitting the data to a 1:1 binding isotherm (inset, solid line).

### 3.3. Preparation of L$^{CN}$ and the Associated Cage Cd•H$^{CN}$; Crystal Structure of Cd•H$^{CN}$

The same new synthetic methodology has allowed the preparation of the nitrile-functionalised ligand **L$^{CN}$**, with the substituents again at the pyridyl C⁴ position, and its use to the prepared octanuclear cage **Cd•H$^{CN}$** which contains 24 externally-directed nitrile groups. For this, the starting material was the known compound 4-cyanopyridine (Scheme 1) [38]. Apart from the convenience of introducing the 2-acetyl group in one step, the previous three-step synthetic route [29] would not have worked here as the final step was a Grignard reaction of 2-cyanopyridine with MeMgI to generate a 2-acetyl group, which would clearly be a challenge with another nitrile group present at the 4-position. 2-Acetyl-4-cyanopyridine could be converted by the standard sequence of reactions (Supplementary Materials, p. 9–11) to **L$^{CN}$** which formed the octanuclear cage **Cd•H$^{CN}$** by reaction with Cd(BF₄)₂ (Supplementary Materials, p. 12); **Cd•H$^{CN}$** is soluble only in polar organic solvents such as MeCN and DMF. Again, high-resolution ES mass spectrometry confirmed the formulation of the cage. The potential value of the nitrile functional groups is that they provide a site for facile further functionalisation in many ways, such as reduction to an amine and then conjugation to a peptide; coordination to additional metal ions to crosslink cages; or a Huisgen-type 'click' cycloaddition reaction with an azide to give a set of 24 units of any desired external substituent around the cage connected via tetrazole spacers [47–49].

Views of the crystal structure of the Cd(II) cage **Cd•H$^{CN}$** are shown in Figure 3. The basic structure of the cage core is not affected by the presence of the nitrile substituents, and consequently, it has the same basic architecture as the parent cage **H** with a Cd(II) ion at each vertex of an approximate cube, and a bridging ligand spanning each edge [35]. The whole assembly is centrosymmetric, with one half of the complex (four metal ions and six ligands) in the asymmetric unit, and one complete molecule in the *P*–1 unit cell.

A diagonally opposite pair of Cd(II) ions [Cd(2) and its symmetry equivalent] have a *fac* tris-chelate coordination geometry, with the remaining six Cd(II) ions all having a *mer* tris-chelate geometry, such that there is a (non-crystallographic) threefold axis passing through Cd(6) and Cd(6′) in addition to the inversion centre, meaning that the complex as a whole has $S_6$ molecular symmetry [35].

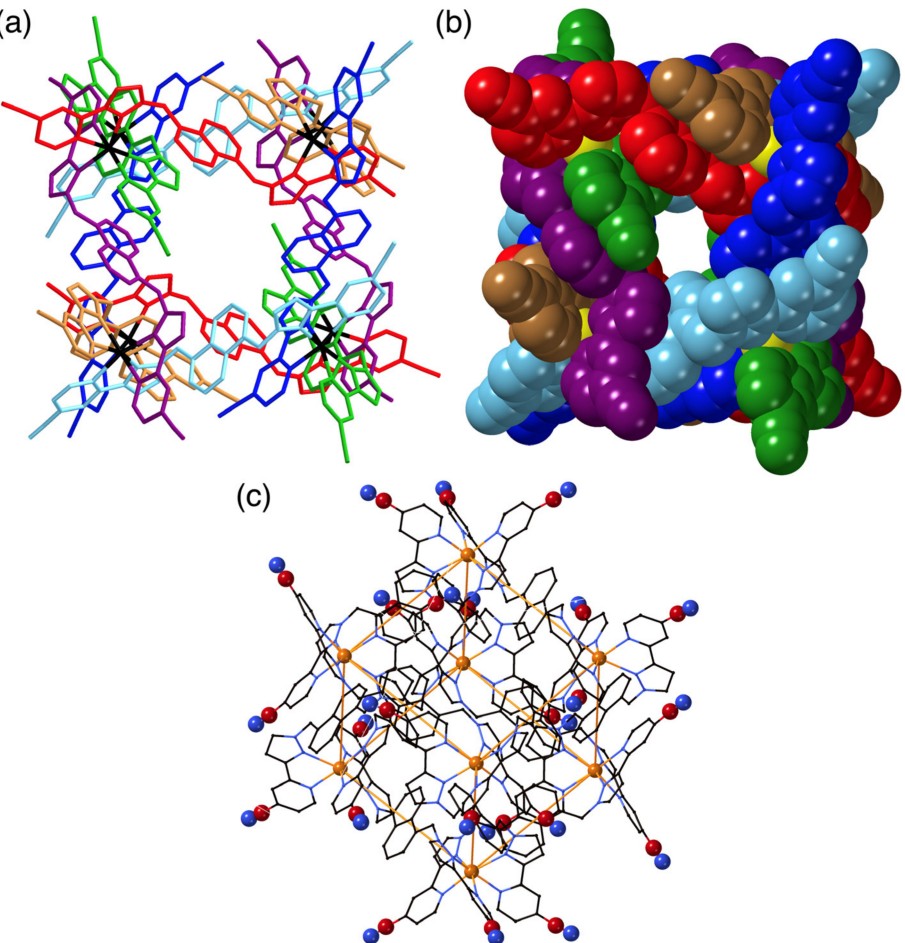

**Figure 3.** Three views of the crystal structure of the Cd(II) cage complex **Cd●H$^{CN}$**. (**a**) A wireframe view with each ligand coloured separately for clarity; (**b**) a space-filling view with the cage in the same orientation, emphasising the intertwined nature of the ligands; and (**c**) a view with most of the cage in wireframe but the CN functional groups shown with larger spheres (C, red; N, blue), emphasising the pseudo-spherical array of 24 externally-directed CN groups.

The flexibility of the ligands allows them to bend at the $CH_2$ groups which act like hinges, meaning that the ligands can adopt conformations in which inter-ligand stacking interactions between electron-rich naphthyl groups and electron-deficient pyrazolyl-pyridine units (coordinated to $Cd^{2+}$ metal ions) generates multi-component pi-stacks around the cage exterior. This occurs also in **H** [35] and **H$^w$** [29], but we can see here how some of the nitrile groups—those which are not externally directed, away from the cage core—participate in the stacking interactions (a detail of one of the 5-component π-stacks is in Figure 4a), in which separation between mean planes of adjacent overlapping aromatic ligand fragments is in the typical 3.3–3.5 Å range. A view of the cage with the nitrile groups emphasising how they form an externally directed pseudo-spherical array is included in Figure 3c. Extensive disorder of the lattice solvent molecules meant that the diffuse electron density associated with them could not be modelled and had to be removed from the refinement using a solvent mask facility.

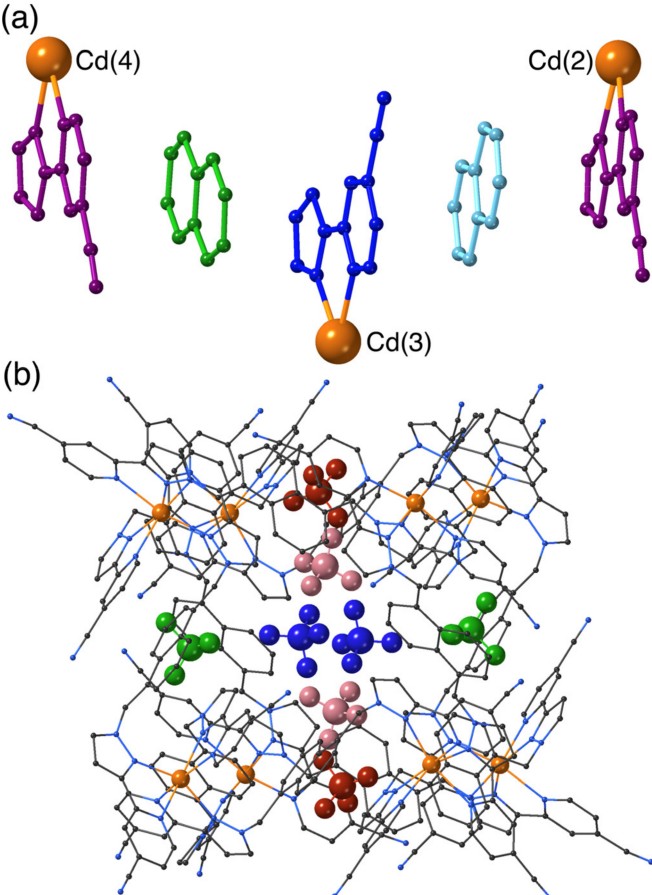

**Figure 4.** Additional views of the crystal structure of the Cd(II) cage complex **Cd•H$^{CN}$**: (**a**) a view of a five-component π-stack between alternating electron-rich and electron-poor ligand fragments; (**b**) wireframe view of the cage showing the location of some of the anions, with symmetry-equivalent pairs coloured the same, embedded in the portals in the cage surface.

The location of the anions is also of interest, as it is the accumulation of anions around the cationic cage surface in the solution that drives the catalysis that we have observed, in particular the cage-catalysed Kemp-elimination in which cavity-bound benzisoazole reacts extremely fast with adjacent surface-bound hydroxide ions [36]. We have previously shown that the windows in the centres of the faces of the cubic assembly provide recognition and binding sites for a wide range of anions via a combination of charge-assisted hydrogen-bonding and hydrophobic effects [50]. We can see that some tetrafluoroborate anions in the structure of **Cd•H$^{CN}$** are associated with the cage surface in this way, though disorder of some of them precludes detailed structural analysis. Two of the cage faces (a crystallographically equivalent pair, opposite each other) contain a fluoroborate anion in a single position with 100% occupancy, this is the anion containing B(21X), coloured blue in Figure 4b. Another pair of faces bind the anion containing B(31X), coloured green in Figure 4b, which has a site occupancy of 0.5; the remaining two faces bind the anion containing B(51X) (dark red in the figure) which also has a site occupancy of 50% in this position; there is also a position close to these anions based on B(51Y), with 25% site occupancy, which lie closer into the centre of the cavity than the B(51X) tetrafluoroborate anion. In all cases, multiple CH•••F interactions occur with the ligands in the cage superstructure with H•••F contacts around 2.5 Å.

### 3.4. Preparation of L$^{CC}$ and the Associated Cage H$^{CC}$; Crystal Structure of Ni•H$^{CC}$

In addition to the new synthetic methodology—the one-step introduction of the 2-acetyl group onto the pyridine nucleus, outlined above—which has allowed the preparation

of ligand $L^{PEG}$ and $L^{CN}$ and their associated cages, we can also functionalise the ligands by simple modifications of existing external functional groups. As part of a general strategy for the external functionalisation of cages, of which the work in this paper is a part, we investigated alkylation of the hydroxyl groups of $L^{W}$ [29] as a means to introduce more versatile and generally synthetically useful substituents, *viz.* alkynyl groups which can also be used for further functionalisation via the well-known CuAAC "click" coupling reaction with azides [51,52]. To this end reaction with pre-formed $L^{W}$ with propargyl bromide in the presence of base allowed the straightforward attachment of two terminal alkyne groups to the pyridine rings ligand core, attached by a short flexible chain, to give the new ligand $L^{CC}$ (Scheme 2; Supplementary Materials, p. 13). To confirm that attachment of these external substituents does not impede cage assembly we used $L^{CC}$ to prepare the $Ni^{2+}$ cage $Ni \bullet H^{CC}$ following the usual methodology (Supplementary Materials, p. 14); it is sparingly soluble in MeCN and soluble in DMF. Mass spectrometric evidence confirmed the formulation of the complex. Note that the paramagnetism of octahedral Ni(II) complexes broadens their $^1$H NMR spectra to the point of being of no value so a $^1$H NMR spectrum for $Ni \bullet H^{CC}$ is not included in the Supplementary Materials. However crystallographic analysis confirms that the Ni(II) cage $Ni \bullet H^{CC}$ has a core structure similar to what we observed with $Cd \bullet H^{CN}$, but bearing 24 alkyne groups on the external surface.

**Scheme 2.** Alkylation of the pendant HO groups of $L^{W}$ (propargyl bromide, NaH, thf) to give $L^{CC}$.

The crystal structure of $Ni \bullet H^{CC}$ is shown in Figure 5. The main features of the $M_8L_{12}$ cage core (arrangement of metal ions and ligands, symmetry, inter-ligand π-stacking, the interaction of anions with the surface portals and so on) are basically the same as in $H^{CN}$, reported above, and do not need re-explaining. The main point however is that the 24 alkyne groups with which the exterior surface is decorated do not interfere with cage assembly and clearly provide a platform for further attachment of a very wide range of substituents.

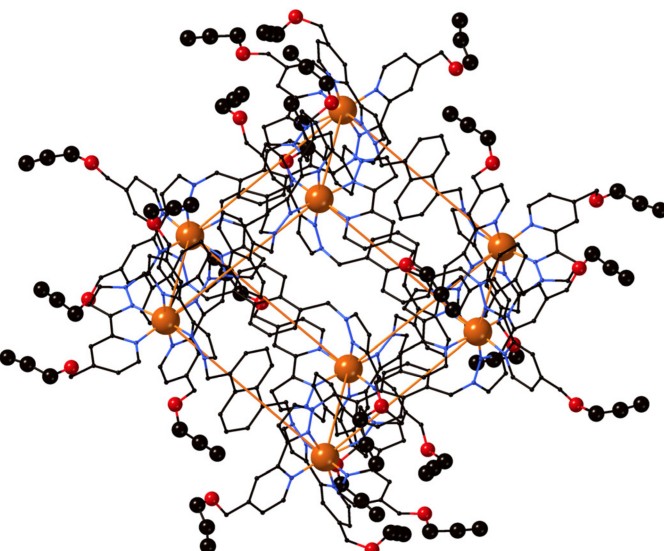

**Figure 5.** Crystal structure of the Ni(II) cage complex **Ni●H^CC**. The core structure is the same as those shown in Figures 1 and 3, and is shown in wireframe; the pendant –OCH$_2$–CCH groups are shown with their atoms as larger spheres (O, red; C, black) to emphasise the pseudo-spherical array of 24 externally-directed terminal alkyne groups.

## 4. Discussion

Overall, the ability to provide a range of externally-directed functional groups around the cage surface is an important development as this strongly influences how the cage hosts interact with the outside world: it therefore provides quite a different research focus from the inward-looking aspects of host/guest complex formation based on the central cavity. At a simple level external substituents control solubility, which is of fundamental importance for optimising host/guest complex formation which is highly solvent-dependent, and we can see for **Co●H^PEG** how the high solubility in CHCl$_3$ allows for much stronger guest binding (by hydrogen-bonding to the interior surface) than occurs in more polar solvents which were previously necessary to dissolve members of this cage family. At a more sophisticated level, the attachment of appropriate external groups could allow binding to surfaces or recognition by biomolecules, and the nitrile or alkyne functional groups which we have appended to the cage exterior surface will be particularly versatile in this respect. A tempting target, for example, is the attachment of glycans to facilitate recognition by lectins: an array of appropriate glycans on the cage exterior could result in strong binding to, for example, pathogenic proteins such as the cholera toxin or could allow cell ingress via recognition routes involving glycans. The previously-demonstrated ability of this cage to bind drug molecules in its cavity [53] could thereby provide a mechanism for delivery of a drug molecule to a specific target which is recognised via interactions with the cage exterior surface, thereby combining different recognition processes associated with the cage interior, and the exterior functional groups, working together.

**Supplementary Materials:** The following are available online at https://www.mdpi.com/article/10.3390/chemistry3040088/s1. Crystallographic CIF and CIF-check files; full details of synthesis and spectroscopic characterisation of new compounds.

**Author Contributions:** Synthesis and characterisation of new ligands and cages: G.D.J., M.B.T., C.P., C.G.P.T. and J.R.P.; X-ray crystallography: C.G.P.T.; guest binding measurements, C.M.; project conception, supervision and manuscript preparation: M.D.W. All authors have read and agreed to the published version of the manuscript.

**Funding:** This research was funded by The Leverhulme Trust (grant number RPG-2019-149); The European Union (H2020-MSCA-ITN grant 'NOAH', project ref. 765297); EPSRC (grant number

EP/R03382X/1); and the University of Warwick. We thank also the Diamond Light Source for the beamtime (proposal MT19876) and the staff of beamline I-19 for assistance.

**Institutional Review Board Statement:** Not applicable.

**Informed Consent Statement:** Not applicable.

**Data Availability Statement:** Data underpinning the work in this paper that is not already included in Supplementary Materials is available on request from the corresponding author.

**Conflicts of Interest:** The authors declare no conflict of interest. The funders had no role in the design of the study; in the collection, analyses, or interpretation of data; in the writing of the manuscript, or in the decision to publish the results.

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
