# Peer review of "A Family of Externally-Functionalised Coordination Cages"

_chemistry, doi:10.3390/chemistry3040088_

Round 1

Reviewer 1 Report

I am hereby providing a review of the article titled “A family of externally-functionalised coordination cages” by Garrett D. Jackson et al. The article describes the syntheses of new ligands (displaying lateral tails able to improve solubility as well as enable further derivatization in the complexity of the systems) and their coordination chemistry towards M2+ metals like Co, Ni and Cd to form [M8L8]16+ cages that are very interesting, since they can encapsulate counterions and other guests. Moreover, the modification of the “outside” of the cage is a nice achievement for the future perspective of improving solubility in aqueous (and non) media as well as controlling further structural growth.

The importance of this manuscript is certainly not to be underestimated in the field of supramolecular chemistry and the manuscript seems to be well-written. Thus I am inclined to accept the manuscript for publication. I am asking the authors to introduce minor corrections/modifications:

  • The abstract is too general, in my opinion. The authors should indicate clearly the metals they used for the complexation and clarify which ligand is used for which metal. After reading the abstract I had the sensation that a large pool of M2+ metals were used in this article, but then I learnt only Co2+, Ni2+ and Cd2+ were used (and not for a full screening with all the synthesized ligands).
  • To my knowledge, LPEG is not indicated in the abstract, yet it is mentioned in the text. It should also be mentioned in the abstract;
  • I think the authors should indicate in the main manuscript the number of the pages describing the ESI (all the references like those in Lines 125, 135, 250, for instance) in SI, so it will be easier to locate the data they are talking about within SI.
  • The authors state that the improved solubility of Co.HPEG complex in CHCl3 allow for stronger binding of guests in the cavity via hydrogen bonds. They calculated the constants in CH3Cl and in CH3CN and their hypothesis is confirmed. This is all very nice. I suggest the author to add a simple scheme in Figure 2 (maybe in the right part of the figure, since there is space), drawing the molecule of coumarin and the postulated hydrogen interactions with the cage (just a sketch, to make readers understand visually the molecules involved in the binding). Clearly the authors must state in the caption that the interactions are postulated, in absence of clear X-ray/computational data.
  • At line 208 the authors mention methanol. I believe it is coming from the synthesis rather than the crystallization. Why is it still present? Is it impossible to eliminate it under heavy vacuum? If I am not wrong I have seen quite substantial residual solvent contaminations (like CH3CN, H2O and MeOH) in the NMR as well. Is it so difficult to remove those solvents under high vacuum for longer time and/or under mild heat? I would like the authors to comment on this point or put cleaner NMR spectra in SI.
  • What catalysis are the authors talking about in Lines 217-219? They added the reference, alright, but I think the understanding of the text will improve dramatically if the authors will add a small line about such catalysis.
  • I am a bit confused about this sentence: “The windows in the centres of the faces of the cubic assembly provide recognition and binding sites for a wide range of anions via a combination of charge-assisted hydrogen-bonding and hydrophobic effects”. The authors mention a wide range of anions, but I only see BF4-, treated here. Are they referring again to reference 36? If so, they should add the reference. I am not familiar with this terminology: "charge-assisted hydrogen bonding". Could the author provide me an explanation for such bond based on topology?
  • I do not seem to find NiLCC NMR spectrum anywhere in SI. Why is that? I understand that NMR techniques (at least monodimensional NMRs cannot provide crucial information on these systems), but I think the authors should put one in SI, as they did for the other complex.

Typos:

  • Line 66: “shwoing” for “showing”.
  • Line 126:”foe” for “for”.
  • Line 250: “he” for “the”.

Suggestions for future work:

  • I suggest the authors to seek a theoretical/computational collaboration (if they do not have done it yet) and start publishing their future manuscripts on this interesting chemistry as a combo of experimental/theoretical papers. The collaboration will improve systematically and profoundly the understanding of covalent and (mostly) non-covalent interactions in these cages from a topological point of view.

Author Response

Referee's comment: 

  • The abstract is too general, in my opinion. The authors should indicate clearly the metals they used for the complexation and clarify which ligand is used for which metal. After reading the abstract I had the sensation that a large pool of M2+ metals were used in this article, but then I learnt only Co2+, Ni2+ and Cd2+ were used (and not for a full screening with all the synthesized ligands).

Response: details of the metal ions used are included in the abstract.

Referee's comment: 

  • To my knowledge, LPEG is not indicated in the abstract, yet it is mentioned in the text. It should also be mentioned in the abstract;

Response: mention of Lpeg is included in the abstract, and teh other ligand abbreviations are included too.

Referee's comment:

  • I think the authors should indicate in the main manuscript the number of the pages describing the ESI (all the references like those in Lines 125, 135, 250, for instance) in SI, so it will be easier to locate the data they are talking about within SI.

Page numbers are now included in the references to specific syntheses in ESI.

Referee's comment:

  • I suggest the author to add a simple scheme in Figure 2 (maybe in the right part of the figure, since there is space), drawing the molecule of coumarin and the postulated hydrogen interactions with the cage (just a sketch, to make readers understand visually the molecules involved in the binding). Clearly the authors must state in the caption that the interactions are postulated, in absence of clear X-ray/computational data.

This has been published before and is known (including X-ray crystal structures); all that is different here is the effect of solvent on the binding constants.  A reference to the earlier work analysing hydrogen-bonding interactions is already present (ref. 37).  A comment about X-ray crystal structures which further confirm the exact nature of the hydrogen bonds has been included (revised text is bottom of page 4).  There is no need for an extra figure.

Referee's comment:

  • At line 208 the authors mention methanol. I believe it is coming from the synthesis rather than the crystallization. Why is it still present? Is it impossible to eliminate it under heavy vacuum? If I am not wrong I have seen quite substantial residual solvent contaminations (like CH3CN, H2O and MeOH) in the NMR as well. Is it so difficult to remove those solvents under high vacuum for longer time and/or under mild heat? I would like the authors to comment on this point or put cleaner NMR spectra in SI.

This comment in line 208 was an error.  The crystal structure does not include MeOH, it includes (substantially disordered) MeCN.  The comment has been adjusted accordingly.  Yes, there are traces of MeOH from the synthesis in the NMR sample, it is difficult to remove completely and is always present when a solvothermal synthesis in MeOH is carried out.

Referee's comment:

  • What catalysis are the authors talking about in Lines 217-219? They added the reference, alright, but I think the understanding of the text will improve dramatically if the authors will add a small line about such catalysis.

Done - a comment is added.

Referee's comment:

  • I am a bit confused about this sentence: “The windows in the centres of the faces of the cubic assembly provide recognition and binding sites for a wide range of anions via a combination of charge-assisted hydrogen-bonding and hydrophobic effects”. The authors mention a wide range of anions, but I only see BF4-, treated here. Are they referring again to reference 36? If so, they should add the reference.

Response: This refers to earlier work; I agree the text was not clear.  The text has been adjusted to make this clear, and the reference (which had not been included, but should have been) has now been added - this is now ref. 50.  So the distinction between what was covered in earlier work and what is new here is now clear.

Referee's comment:

  • I am not familiar with this terminology: "charge-assisted hydrogen bonding". Could the author provide me an explanation for such bond based on topology?

The phrase 'charge-assisted hydrogen bonding' is extremely standard and well known in the field and does not need re-defining.

Referee's comment: 

  • I do not seem to find NiLCC NMR spectrum anywhere in SI. Why is that? I understand that NMR techniques (at least monodimensional NMRs cannot provide crucial information on these systems), but I think the authors should put one in SI, as they did for the other complex

The paramagnetism of Ni(II) results in NMR spectra that are so broad as to be completely useless.  Whilst some paramagnetic ions like high-spin Co(II) spread 1H signals out over a wide chemical shift range and act as useful shift reagents, octahedral Ni(II) does not do this and simply broadens spectra so that no useful information can be discerned.  A comment has been added concerning this.

Referee's comment

  • Line 66: “shwoing” for “showing”.
  • Line 126:”foe” for “for”.
  • Line 250: “he” for “the”

Response: all fixed

Referee's comment:

  • I suggest the authors to seek a theoretical/computational collaboration (if they do not have done it yet) and start publishing their future manuscripts on this interesting chemistry as a combo of experimental/theoretical papers. The collaboration will improve systematically and profoundly the understanding of covalent and (mostly) non-covalent interactions in these cages from a topological point of view

Response: the reviewer should check our previous published work.  No changes are required to the manuscript in respect of this.

Reviewer 2 Report

The authors develop new synthetic routes for M8L12 cubic coordination cage with different substitutes. These materials have potential applications in transport and release of cargoes. The result is very interesting but the innovation needs to be strengthened comparing with the previous report (Chem. Sci. 2013, 4, 2744-2751). Therefore, I suggest that it can be published in Chemistry after major revision. The following aspects need to be considered for improvement.

  1. This manuscript is focus on the synthesis of Cd HCN and Ni HCC. Please present the 1H NMR spectrum of Cd HCN and Ni HCC to confirm the chemical states.
  2. The synthesis of LPEG and LCN a key step in the preparation of coordination cage, the yield and pure of LPEG and LCN should be provided in the manuscript.
  3. The physical-chemical properties of as prepared coordination cages, especial the solubility of in water and organic solvents, should be investigated.

Author Response

Referee's comment:  This manuscript is focus on the synthesis of Cd HCN and Ni HCC. Please present the 1H NMR spectrum of Cd HCN and Ni HCC to confirm the chemical states.

Response: For Cd HCN the 1H NMR spectrum is already included in ESI.  For Ni HCC, see comments addressed to review 1 who made the same point.  the paramagnetism of octahedral Ni(II) makes the 1H NMR spectra so broadened as to be useless

Referee's comment:  The synthesis of LPEG and LCN a key step in the preparation of coordination cage, the yield and pure of LPEG and LCN should be provided in the manuscript.

Response: The yields are included with the synthetic details in the ESI. There is no particular need to repeat them in the main text of the manuscript, the point of the ESI is that routine experimental information does not need to be in the main text.

Referee's comment:  The physical-chemical properties of as prepared coordination cages, especial the solubility of in water and organic solvents, should be investigated.

Response: comments on the solubility have been added to the main text where appropriate, alongside the discussion of synthesis and characterisation, as this is indeed an important issue.

Reviewer 3 Report

This manuscript reports syntheses of some new cage molecules along with two crystal structures. The synthetic work seems to be well conducted, even if the ligand synthes appear rather tedious. The use og PEG to make complexes, which are soluble in water as well as in non-polar solvents looks a good idea.

The R indexes for X-ray structures look very high, but they are typical for large systems, which inglude poorly ordered solvent molecules and counter ions. This is well explained in the manuscript and CIFs.

The experimental part is lacking elemental analyses but the spectroscopic characterisations support the idea on single components. 

The metals M, should be given in abstract. Moreover, check the initials for professor Sharpless.

In general, the manuscript looks good and can be published in this journal.

Author Response

This reviewer has only very inor suggestions as follows:

Comments; The metals M, should be given in abstract. Moreover, check the initials for professor Sharpless.

Response. the metals M are now given in the abstract (reviewer 1 also asked for this).

Round 2

Reviewer 2 Report

The revised manuscript can be accepted and published on Chemistry.